# Knowledge and attitudes of medical students towards the health impact of climate change: A study from Jordan

Mohammed O. Khabour[1], Owais Omar Tarabsheh[1], Bilal M. Al-zu'bi[1], Omar F. Khabour[2], Rami Saadeh[1]*

1 Faculty of Medicine, Jordan University of Science and Technology, Irbid, Jordan, 2 Department of Medical Laboratory Sciences, Jordan University of Science and Technology, Irbid, Jordan

* rasaadeh@just.edu.jo

## Abstract

Studies have reported a strong relationship between climate change and human health. Medical students' knowledge and attitudes toward the impact of climate change on health are crucial to fostering their environmental stewardship. Therefore, the aim of this study was to examine the awareness and attitudes of medical students in Jordan toward climate change and human health. The study was cross-sectional in design, anonymous, self-reported, and used a closed-ended questionnaire. The study included 837 students from various medical specialties, including medicine, dentistry, applied medical sciences, pharmacy, and nursing. Statistical analysis involved cross-tabulations and regression analysis. About 46.3% of students reported good awareness of the health impacts of climate change, while 44.8% reported somewhat awareness. This awareness was found to be associated with female gender (P = 0.003) and university level (P < 0.001). In addition, students showed a positive attitude toward the importance of climate change to human health (attitude score = 19.7 out of 24), including the integration of climate change into university curricula. The internet (88.9%) and social media (86.5) were the major sources of information reported by students about climate change. Climate change related illnesses reported by students included air quality/respiratory illnesses, extreme weather-related illnesses, infectious disease outbreaks, physical inactivity, and mental health. In conclusion, medical students in Jordan have an acceptable level of knowledge and positive attitudes toward climate change. This could be improved through interventions that integrate climate change into university curricula.

## Introduction

Climate change is a global concern that negatively affects people's lives [1,2]. Phenomena associated with climate change are plenty and include extreme weather (heat waves, cold waves, and floods), loss of glacier mass, the greenhouse effect,

**Data availability statement:** All relevant data are within the paper and its Supporting Information files.

**Funding:** The study was supported by funds from Jordan University of Science and Technology (fund number: 210-2024) and the GeoHealth Hub for Climate Change and Health in the Middle East and North Africa (Fogarty NIH awards number 5U2RTW012228 and U01TW012237). The funders had no role in study design, data collection and analysis, decision to publish, or preparation of the manuscript.

**Competing interests:** The authors have declared that no competing interests exist.

and sea level rise [3,4]. Climate change occurs as a result of natural causes, such as volcanoes and fluctuating solar emissions, and human activities, such as reliance on fossil fuels for energy production, forest destruction, and pollution [5–7]. The negative impacts of climate change encompass many aspects. For example, climate change is associated with disruptions to water and food supplies, loss of resources, and migration of people [8–11]. These impacts have detrimental effects on health, such as the spread of malnutrition, waterborne/airborne- related illnesses, transmission of infectious diseases, changes in lifestyles (e.g., sleep patterns, time allocation, physical activity), and mental illnesses [12–16]. As part of addressing the impacts of climate change on health, enhancing students' awareness of climate change in medical colleges is crucial [17,18]. Integrating climate change into university curricula can equip students with the knowledge needed to interact positively with their environment and climate change [19–21].

Although numerous studies have been conducted on the health impacts of climate change, these impacts have not been adequately considered in public health discussions in the Middle East and North Africa (MENA). In addition, investigations examining the attitude and awareness of students in medical fields about the health consequences of climate change in the MENA are limited [22,23]. For example, studies of students from nursing colleges in MENA have highlighted the need to integrate climate change into nursing academic programs [24–26]. Medical students represent the future workforce and leaders of the health system and should be equipped with adequate knowledge and expertise to address the impact of climate change on human health. Moreover, medical students can be influential and advocates for climate health among student communities and the public. Therefore, it is essential to ensure that medical students have a strong awareness and positive attitude toward climate change and health. The current research aimed to examine the knowledge and attitudes of students from various medical fields towards climate change and its impact on health, including the integration of climate issues into university curricula.

## Methods

### Study design and target population

This study aimed to examine students' knowledge and attitudes toward the impact of climate change on human health. The study was cross-sectional in design and questionnaire based. The study targeted students (age ≥ 18 years) enrolled in medical colleges at Jordanian universities (medicine, applied medical sciences, dentistry, nursing and pharmacy). Students from schools other than the medical field or studying abroad were excluded from this study.

### Data collection

Recruitment was performed during the period of April 1st, 2024, to August 1st, 2024. Participation in the study was announced through student groups on various social media platforms, which include Facebook, WhatsApp, Instagram, and others. Sample size was determined using GPower-3.1 software, with an effect size of 0.1, an alpha of 0.05, and a power of 0.9. The calculated sample size was 850 participants.

## Ethical approval

Ethical approval to conduct the study was granted from the Research Ethics Committee of King Abdullah University Hospital (ID: 2024/167/24). The first part of the questionnaire contained a detailed description of the study objectives and procedures. Electronic consent was granted from participants before filling out the questionnaire. No personal information was collected, and the questionnaire was anonymous.

## Study instrument

The "Google Forms" tool was adopted to upload and deliver the study instrument to the target population. The study instrument included several questions to collect demographic information about the study sample, such as age, academic major, gender, university level, and income. Participants were asked to report their level of awareness of the health impacts of climate change. Students were given three options to choose from (aware to a good extent, somewhat aware, not aware). Students were then asked about the potential impacts of climate change on health. They were presented with 15 potential impacts with "yes or no" options to choose from. Students' attitudes toward the health impacts of climate change were measured using 8 items with three choices ("agree = points, neutral = 2 points, disagree = 1 point") presented to students to choose from. Attitude scores were calculated out of 24. The sources used by students to gain knowledge about climate change and health were also assessed in the instrument using a multiple-choice grid question that included items such as the internet, social media, university course, (etc.), with "yes or no" options. The instrument questions were adopted from previous similar studies [26–30]. Before distributing the questionnaire, it was reviewed by two experts in the field to ensure its clarity, relevance, and content accuracy. The questionnaire was validated by pilot testing on 20 participants. The questionnaire questions were revised accordingly.

## Statistical analysis

SPSS software was employed for statistical testing. These include frequencies, cross-tabulations and regression analysis. A P-value<0.05 was chosen to infer statistical differences. Study data are available as supplementary file (Sf1).

## Results

In the current study, a total of 873 students were included in the study. The mean age±standard deviation was 21.4±1.63 years (age range 18–26 years). The percentages of male and female students were comparable (47.1% and 52.8%, respectively). Most students were from the College of Medicine (71.2%), Jordanian (87.6%), and living in a city (64.0%). The distribution of family income categories was similar (Table 1). More than one-third of the students were in their fourth year of study (Table 1).

The results indicate that 46.3% and 44.8% reported having good and somewhat awareness of the health impacts of climate change, respectively. In addition, 8.9% reported a lack of awareness of the health impacts of climate change.

Table 2 shows cross-tabulations of self-reported awareness of the impact of climate change on health with participants' demographics. Female students reported significantly higher awareness than male students (P=0.003). In addition, university level (number of years in the program) was significantly associated with students' awareness of the impact of climate change on health (P<0.001). In general, students' awareness increases with university level (Table 2). However, factors such as students' discipline, nationality, income, and place of living were not associated with students' awareness of the impact of climate change on health (P>0.05). Logistic regression of variables associated with Students' awareness of the impact of climate change on health is shown in Table 3. Both gender and university level were found to be related to students' awareness (P<0.01). As in cross-tabulations, students' discipline, nationality, income, and place of living were not associated with students' awareness of the impact of climate change on health (P>0.05).

Table 4 shows students' attitudes toward the health aspects of climate change. Many students expressed positive attitudes toward the impact of climate change on human health, with their attitude score of 19.7 out of 24. Students agreed

**Table 1. Demographics of the student participants.**

| Item | Sub-item | Frequency | Percent |
|------|----------|-----------|---------|
| Age (years) | Less or = 21 | 450 | 51.5 |
| | More than 21 | 423 | 48.5 |
| Gender | Male | 411 | 47.1 |
| | Female | 462 | 52.9 |
| University discipline | Medicine | 622 | 71.2 |
| | Dentistry | 31 | 3.6 |
| | Nursing | 110 | 12.6 |
| | Pharmacy | 45 | 5.2 |
| | Applied Medical Sciences | 65 | 7.4 |
| Income | Less than 700 JD | 252 | 28.9 |
| | 700-1500 JD | 347 | 39.7 |
| | > 1500 JD | 274 | 31.4 |
| Nationality | Jordanian | 765 | 87.6 |
| | Non-Jordanian | 108 | 12.4 |
| Place of living | City | 559 | 64.0 |
| | Village | 314 | 36.0 |
| University Level | First/second year | 190 | 21.8 |
| | Third year | 177 | 20.3 |
| | Fourth year | 363 | 41.6 |
| | Fifth/Sixth year | 143 | 16.4 |

that climate change impacts health and this issue should be included in university curricula. However, only 37.9% agreed that climate change is controllable.

Table 5 shows the sources students used to gain knowledge about the impact of climate change on health. The internet and social media were the major sources. Minor reported sources included scientific journals, university courses, and friends.

Students were asked about the potential health impacts of climate change (Table 6). More than 80% agreed that climate change is associated with health conditions such as air quality-related illnesses, heat/cold related illnesses, water/vector-borne infectious diseases, and respiratory diseases. Students scored lower on the impact of climate change on conditions such as obesity (58.6%), mental health (68), cardiovascular diseases (65.8%) and malnutrition (73.3%). Finally, 79% agreed that extreme weather could disrupt health services (Table 6).

## Discussion

In the current study, students' awareness and attitudes towards health impacts of climate change were investigated. The results showed that most students reported being aware of the health impacts of climate change. This finding is consistent with several previous studies A study from China reported that 83% of medical, public health and nursing students had good awareness of the health impacts of climate change [28]. A qualitative study of medical students from Iran showed that medical students had a good understanding of the impact of climate change on health and society [31]. Good awareness of health issues associated with climate change among medical students has been reported in studies conducted in India and Ethiopia [29,30,32].

The results of the present study showed that medical students had positive attitudes toward climate change and human health, including the integration of climate change into university curricula. A study conducted in five European countries also reported positive attitudes among nursing students towards the integration of climate change and health into

**Table 2. Awareness of participants of the impact of climate change on health.**

| Item | Sub-item | Not aware N(%) | Aware to some extent N(%) | Aware to a good extent N(%) | P-value |
|------|----------|----------------|---------------------------|-----------------------------|---------|
| Gender | Male | 51(12.4) | 185(45.0) | 175(42.6) | 0.003 |
| | Female | 27(5.8) | 219(47.4) | 216(46.8) | |
| University discipline | Medicine | 48(7.7) | 297(47.7) | 277(44.5) | 0.172 |
| | Dentistry | 4(12.9) | 12(38.7) | 15(48.4) | |
| | Nursing | 13(11.8) | 37(33.6) | 60(54.5) | |
| | Pharmacy | 4(8.9) | 20(44.4) | 21(46.7) | |
| | Applied Medical Sciences | 9(13.8) | 25(38.5) | 31(47.7) | |
| Income | Less than 700 JD | 28(11.1) | 117(46.4) | 107(42.5) | 0.327 |
| | 700-1500 JD | 30(8.6) | 151(43.5) | 166(47.8) | |
| | > 1500 JD | 20(7.3) | 136(49.6) | 118(43.1) | |
| Nationality | Jordanian | 68(8.9) | 344(45.0) | 353(46.1) | 0.088 |
| | Non-Jordanian | 10(9.3) | 60(55.6) | 38(35.2) | |
| Place of living | City | 55 (9.8) | 252(45.1) | 252(45.1) | 0.375 |
| | Village | 23(7.3) | 152(48.4) | 139(44.3) | |
| University Level | First/second year | 33(17.4) | 84(44.2) | 73(38.4) | 0.000 |
| | Third year | 18(10.2) | 76(42.9) | 83(46.9) | |
| | Fourth year | 20(5.5) | 170(46.8) | 173(47.7) | |
| | Fifth/Sixth year | 7(4.9) | 61(42.7) | 75(52.4) | |

**Table 3. Logistic regression of factors that are associated with awareness of participants of the impact of climate change on health.**

| Item | B | Standard Error | Wald | Degree of freedom | P-value | Odds ratio | 95% confidence intervals (Lower-Upper) |
|------|---|----------------|------|-------------------|---------|------------|----------------------------------------|
| Student discipline | | | 2.307 | 4 | 0.679 | | |
| Applied Medical Sciences | -0.173 | 0.663 | 0.068 | 1 | 0.794 | 0.841 | 0.229-3.085 |
| Dentistry | -0.491 | 0.776 | 0.401 | 1 | 0.526 | 0.612 | 0.134-2.797 |
| Medicine | 0.250 | 0.571 | 0.192 | 1 | 0.661 | 1.284 | 0.420-3.929 |
| Nursing | 0.101 | 0.633 | 0.026 | 1 | 0.873 | 1.107 | 0.320-3.826 |
| Income | | | 1.736 | 2 | 0.420 | | |
| Less than 750 JD | -0.132 | 0.320 | 0.171 | 1 | 0.680 | 0.876 | 0.468-1.640 |
| More than 750 JD | -0.426 | 0.343 | 1.544 | 1 | 0.214 | 0.653 | 0.333-1.279 |
| Jordanians | 0.191 | 0.377 | 0.256 | 1 | 0.613 | 1.210 | 0.578-2.533 |
| Living in a city | -0.471 | 0.280 | 2.829 | 1 | 0.093 | 0.624 | 0.360-1.081 |
| Female students | 0.962 | 0.263 | 13.371 | 1 | 0.000 | 2.616 | 1.562-4.381 |
| University level | | | 19.697 | 3 | 0.000 | | |
| Fifth/sixth year | 0.850 | 0.472 | 3.240 | 1 | 0.072 | 2.340 | 0.927-5.904 |
| First/second year | -0.592 | 0.325 | 3.330 | 1 | 0.068 | 0.553 | 0.293-1.045 |
| Fourth year | .668 | 0.349 | 3.666 | 1 | 0.056 | 1.950 | 0.984-3.864 |

university curriculum [27]. Similar findings have been reported on the attitudes of students from different countries toward climate change and the integration of the health impacts of climate change into university curricula [26,33–35]. In the current study, about 37.9% agreed that climate change could be controlled, which is lower than that reported in a study from China (67%) [28]. Therefore, the results of the present study and the results of previous studies indicate a similar attitude towards climate change and its integration into university curricula [36,37].

**Table 4. Attitude of medical students toward health impacts of climate change.**

| Item | Agree | Neutral | Disagree |
|---|---|---|---|
| "Climate change is an important issue for health" | 669(76.6) | 184(21.1) | 20(2.3) |
| "Climate change is an important issue for my specialty" | 519(59.5) | 295(33.8) | 59(6.8) |
| "Issues about climate change should be included in the academic curriculum of my specialty" | 438(50.2) | 328(37.6) | 107(12.3) |
| "Medical students should learn about the impact of climate change when studying their subject" | 520(59.6) | 269(30.8) | 84(9.6) |
| "Concerns about the environment influenced my choice of university". | 293(33.6) | 275(31.5) | 305(34.9) |
| "Climate change is bad for human health" | 579(66.3) | 249(28.5) | 45(5.2) |
| "Climate change is controllable" | 331(37.9) | 374(42.8) | 168(19.2) |
| "People need more information about climate change" | 695(79.6) | 152(17.4) | 26(3.0) |

**Table 5. Sources used by the students to learn about the impact of climate change on health.**

| Item | Yes: N (%) | No: N (%) |
|---|---|---|
| Social media | 755(86.5) | 118(13.5) |
| University courses | 447(51.2) | 426(48.8) |
| News papers | 332(38.0) | 541(62.0) |
| Internet | 776(88.9) | 97(11.1) |
| Friends | 501(57.4) | 372(42.6) |
| Scientific journals | 417(47.8) | 456(52.2) |

**Table 6. Potential health impact of climate change reported by the students.**

| Item | Yes: N (%) | No: N (%) |
|---|---|---|
| Air quality-related illness | 820(93.3) | 53(6.1) |
| Heat-related illness | 704(80.6) | 169(19.4) |
| Disruption of health services by extreme weather | 690(79.0) | 183(21.0) |
| Cold-related illness | 732(83.8) | 141(16.2) |
| Flood-related displacement | 672(77.0) | 201(23.0) |
| Illness related to shortage of water supply | 720(82.5) | 153(17.5) |
| Vector-borne infectious disease | 721(82.6) | 152(17.4) |
| Water-borne infectious disease | 725(83.0) | 148(17.0) |
| Food-borne disease | 687(78.7) | 186(21.3) |
| Mental Health conditions | 598(68.5) | 275(31.5) |
| Malnutrition | 640(73.3) | 233(26.7) |
| Cardiovascular diseases | 574(65.8) | 299(34.2) |
| Respiratory diseases | 769(88.1) | 104(11.9) |
| Obesity | 512(58.6) | 361(41.4) |
| Physical inactivity | 669(76.6) | 204(23.4) |
| Diabetes | 502(57.5) | 371(42.5) |

The results of the study indicated that knowledge of health impacts of climate change among medical students was associated with female gender and university level. In a study conducted in the United States, it was found that women are more aware than men of the health impacts of climate change [38]. An Ethiopian study reported a similar trend, with greater awareness found among female students and university level [30]. Gender differences in the health impacts of climate change were also reported among young citizens in Malaysia [39]. In contrast, in a study conducted in India, male

students were reported to be more aware of the health impact of climate change than female students [29]. Finally, no association between gender and awareness of health impacts of climate change was reported in a study from the Arab region [26]. Thus, based on most previous studies, gender appears to be an influential factor in students' understanding of the health impacts of climate change.

In the current study, the internet and social media were the major sources of information about the impact of climate change on health. This in consistent with studies conducted in India and Ethiopia, which showed that internet and social media were the major sources of information about climate change among students [30,32]. In addition, studies from Italy and France have indicated the importance of social media as a source of information about climate change for students [40,41]. Thus, social media should be utilized as an educational tool for students to enhance their knowledge about the impacts of climate change.

The health impacts of climate change are well-documented. These include the spread of malnutrition, water/air-related illnesses, transmission of infectious disease, changes in lifestyles, and mental illness [12–16]. In the present study, most participants agreed that climate change contributes to a wide range of health conditions such as air quality/respiratory-related illnesses, extreme weather-related illnesses, the spread of infectious diseases, physical inactivity, and mental health. In studies conducted in India, respiratory illnesses and direct physical harm caused by extreme weather events were reported to be among the most important health impacts of climate change [29,32]. In a study conducted in Germany, medical students were reported to be at increased risk of suffering from mental health problems related to climate change [42].

The impact of climate change on physical activity includes disruption of individuals' routine physical activity, especially outdoor activities. In addition, climate change is usually associated with diseases such as viral respiratory infections and asthma that affect individuals' ordinal physical activity [43–45]. Thus, climate change can affect many aspects of an individual's life. Efforts should be made to improve our environment and reduce pollution. Interventions are needed to integrate climate change and health into university curricula.

A noticeable limitation is that data collection relied on self-reporting, which may introduce bias. In addition, most participants were from school of medicine, therefore limiting the perception from other scientific disciplines. Hence, its recommended that a larger sample of students from other medical and health fields be included in future studies.

In conclusion, medical students in Jordan have an acceptable level of knowledge and positive attitudes toward the health impacts of climate change. In addition, gender and university level may influence students' awareness of the health impacts of climate change. Moreover, medical students rely on the internet and social media to gain knowledge about climate change and health. Integrating climate change into university curricula would enhance the positive role medical students play in supporting the environment.

## Supporting information

**S1 File. Raw data file for the study.**
(XLSX)

## Acknowledgments

The authors thank Mr. Ahmed Kofahi, Mr. Ali Hisham, and Mr. Ali Alomar for their efforts in the recruitment of participants.

## Author contributions

**Conceptualization:** Omar F. Khabour, Rami Saadeh.

**Data curation:** Mohammed O. Khabour, Bilal M. Al-zu'bi.

**Formal analysis:** Mohammed O. Khabour, Owais Omar Tarabsheh, Rami Saadeh.

**Investigation:** Mohammed O. Khabour, Owais Omar Tarabsheh, Bilal M. Al-zu'bi, Omar F. Khabour.

**Methodology:** Mohammed O. Khabour.

**Project administration:** Owais Omar Tarabsheh, Omar F. Khabour, Rami Saadeh.

**Software:** Owais Omar Tarabsheh, Bilal M. Al-zu'bi.

**Supervision:** Omar F. Khabour.

**Validation:** Bilal M. Al-zu'bi, Rami Saadeh.

**Writing – original draft:** Omar F. Khabour.

**Writing – review & editing:** Mohammed O. Khabour, Owais Omar Tarabsheh, Bilal M. Al-zu'bi, Rami Saadeh.

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
