## [Decision Letter · Decision Letter 0]

28 Nov 2024

PONE-D-24-44537Knowledge and attitudes of medical students towards the health impact of climate change: a study from JordanPLOS ONE

Dear Dr. Saadeh,

Thank you for submitting your manuscript to PLOS ONE. After careful consideration, we feel that it has merit but does not fully meet PLOS ONE’s publication criteria as it currently stands. Therefore, we invite you to submit a revised version of the manuscript that addresses the points raised during the review process.

We look forward to receiving your revised manuscript.

Kind regards,

Fatma Refaat Ahmed, Ph.D.

Academic Editor

PLOS ONE

Journal Requirements:

[The study was supported by fund from Jordan University of Science and Technology (fund number: 210-2024).].

3. Thank you for stating the following in the Acknowledgments Section of your manuscript: [Authors thank Mr. Ahmed Kofahi, Mr. Ali Hisham, and Mr. Ali Alomar for their efforts in the recruitment of participants. Authors would like to thank the GeoHealth Hub for Climate Change and Health in the Middle East and North Africa (#5U2RTW012228 and #U01TW012237) for its support.]

Please remove any funding-related text from the manuscript and let us know how you would like to update your Funding Statement. Currently, your Funding Statement reads as follows: "The study was supported by fund from Jordan University of Science and Technology (fund number: 210-2024)."

4. In the online submission form, you indicated that your data is available only on request from a third party. Please note that your Data Availability Statement is currently missing the contact details for the third party, such as an email address or a link to where data requests can be made. Please update your statement with the missing information.

Reviewer's Responses to Questions

**Comments to the Author**

1. Is the manuscript technically sound, and do the data support the conclusions?

Reviewer #1: Yes

Reviewer #2: No

2. Has the statistical analysis been performed appropriately and rigorously? 

Reviewer #1: Yes

Reviewer #2: No

3. Have the authors made all data underlying the findings in their manuscript fully available?

Reviewer #1: Yes

Reviewer #2: No

4. Is the manuscript presented in an intelligible fashion and written in standard English?

Reviewer #1: No

Reviewer #2: No

5. Review Comments to the Author

Reviewer #1: The Author did a great job with the methodology and analysis however the discussion needs more work. the discussion lacks a flow that engages the reader and does not enable ease in connecting the dots of the findings.

Reviewer #2: Kindly revise your article thoroughly following the pattern recommended by the PLOS ONE . For Guidance, read

6. PLOS authors have the option to publish the peer review history of their article (what does this mean? ). If published, this will include your full peer review and any attached files.

**Do you want your identity to be public for this peer review?** For information about this choice, including consent withdrawal, please see our Privacy Policy .

Reviewer #1: No

Reviewer #2: No

---

## [Author Response · Author response to Decision Letter 1]

9 Dec 2024

Dear Editor of PLOS-ONE

We are thankful to the reviewers for their thoughtful and detailed review of our manuscript (ONE-D-24-44537) entitled “Knowledge and attitudes of medical students towards the health impact of climate change: a study from Jordan”. All the recommended changes and edits have been incorporated in the revised version of our manuscript. We greatly appreciate the time and effort invested to produce this detailed review. Our point-by-point response is provided below.

Comment

Response

The manuscript was checked to meet PLOS ONE's style requirements

[The study was supported by fund from Jordan University of Science and Technology (fund number: 210-2024).].

Response

We have revised the Financial Disclosure statement as indicated. The revised statement is:

The revised statement is “The study was supported by funds from Jordan University of Science and Technology (fund number: 210-2024) and the GeoHealth Hub for Climate Change and Health in the Middle East and North Africa (Fogarty NIH awards number 5U2RTW012228 and U01TW012237). The funders had no role in study design, data collection and analysis, decision to publish, or preparation of the manuscript.”

Changes to the Financial Disclosure statement were added to the cover letter.

3. Thank you for stating the following in the Acknowledgments Section of your manuscript: [Authors thank Mr. Ahmed Kofahi, Mr. Ali Hisham, and Mr. Ali Alomar for their efforts in the recruitment of participants. Authors would like to thank the GeoHealth Hub for Climate Change and Health in the Middle East and North Africa (#5U2RTW012228 and #U01TW012237) for its support.]

Please remove any funding-related text from the manuscript and let us know how you would like to update your Funding Statement. Currently, your Funding Statement reads as follows: "The study was supported by fund from Jordan University of Science and Technology (fund number: 210-2024)."

Response

The Acknowledgement statement was revised as suggested. In addition, the financial statement was revised as shown in our response to point 2.

4. In the online submission form, you indicated that your data is available only on request from a third party. Please note that your Data Availability Statement is currently missing the contact details for the third party, such as an email address or a link to where data requests can be made. Please update your statement with the missing information.

Response.

The data availability statement was revised to include contact information.

Reviewer comments

Reviewer #1: The Author did a great job with the methodology and analysis however the discussion needs more work. The discussion lacks a flow that engages the reader and does not enable ease in connecting the dots of the findings.

Response

Thank you for your comment. The Discussion section was revised to ensure proper flow and connections between different findings.

Reviewer #2: Kindly revise your article thoroughly following the pattern recommended by the PLOS ONE.

Response

The manuscript was revised to meet PLOS ONE's style and pattern requirements.

Sincerely yours,

Dr. Rami Saadeh

---

## [Decision Letter · Decision Letter 1]

27 Jan 2025

PONE-D-24-44537R1Knowledge and attitudes of medical students towards the health impact of climate change: a study from JordanPLOS ONE

Dear Dr. Saadeh,

Thank you for submitting your manuscript to PLOS ONE. After careful consideration, we feel that it has merit but does not fully meet PLOS ONE’s publication criteria as it currently stands. Therefore, we invite you to submit a revised version of the manuscript that addresses the points raised during the review process.

We look forward to receiving your revised manuscript.

Kind regards,

Fatma Refaat Ahmed, Ph.D.

Academic Editor

PLOS ONE

Journal Requirements:

Reviewers' comments:

Reviewer's Responses to Questions

**Comments to the Author**

1. If the authors have adequately addressed your comments raised in a previous round of review and you feel that this manuscript is now acceptable for publication, you may indicate that here to bypass the “Comments to the Author” section, enter your conflict of interest statement in the “Confidential to Editor” section, and submit your "Accept" recommendation.

Reviewer #2: (No Response)

Reviewer #3: (No Response)

2. Is the manuscript technically sound, and do the data support the conclusions?

Reviewer #2: Partly

Reviewer #3: Yes

3. Has the statistical analysis been performed appropriately and rigorously? 

Reviewer #2: I Don't Know

Reviewer #3: Yes

4. Have the authors made all data underlying the findings in their manuscript fully available?

Reviewer #2: Yes

Reviewer #3: Yes

5. Is the manuscript presented in an intelligible fashion and written in standard English?

Reviewer #2: No

Reviewer #3: No

6. Review Comments to the Author

Reviewer #2: 1.Kindly add continuous line number to the manuscript

2.ABSTRACT: Kindly add about methodology.

3.INTRODUCTION: Kindly Write a strong rationale

4.Study design is mentioned in caption but not written anywhere. Kindly add name of the study design used.

5.Conclusion may kindly be strengthened.

Reviewer #3: The manuscript addresses an important and timely topic, focusing on medical students’ awareness and attitudes toward the health impacts of climate change, particularly in the underrepresented MENA region.The methodology is robust, with a clearly defined study population, ethical approval, and validated data collection tools.The discussion ties the findings to relevant global and regional studies, emphasizing the need for integrating climate change into medical curricula.

However, the same suggestions for improvement is to address typographical and grammatical errors such as in introduction Sentence: "Phenomena associated with climate change are plenty and include extreme weather (heat waves, cold weaves, and floods)..."Issue: "Cold weaves" should be "cold waves." In methods, Sentence: "Before distributing the questionnaire, the instrument was reviewed by two experts in the field to verify its clarity and content."Issue: No grammatical error, but "clarity and content" could be expanded for specificity.Suggestion: "Before distributing the questionnaire, the instrument was reviewed by two experts in the field to verify its clarity, relevance, and content accuracy." In results, Sentence: "The percentages of male and female students were comparable (47.1% and 52.8% respectively)."Issue: A comma is missing after "52.8%."

7. PLOS authors have the option to publish the peer review history of their article (what does this mean? ). If published, this will include your full peer review and any attached files.

**Do you want your identity to be public for this peer review?** For information about this choice, including consent withdrawal, please see our Privacy Policy .

Reviewer #2: No

Reviewer #3: **Yes: ** Zain Ali

---

## [Author Response · Author response to Decision Letter 2]

31 Jan 2025

Dear Editor of PLOS-ONE

We are thankful to the reviewers for their thoughtful and detailed review of our manuscript (ONE-D-24-44537 R1) entitled “Knowledge and attitudes of medical students towards the health impact of climate change: a study from Jordan”. All the recommended changes and edits have been incorporated in the revised version of our manuscript. We greatly appreciate the time and effort invested to produce this detailed review. Our point-by-point response is provided below.

Academic Editor

PLOS ONE

Comment

Journal Requirements:

Response

Thank you. References were checked. All references are now complete and correct. We don’t have retracted papers among the references.

Reviewers' comments:

Reviewer #2:

Comment

1.Kindly add continuous line numbers to the manuscript

Response

Thank you. Continuous line numbers were added to the manuscript.

Comment

2.ABSTRACT: Kindly add about methodology.

Response

The Methodology part was expanded in the Abstract as suggested.

Comment

3.INTRODUCTION: Kindly Write a strong rationale

Response.

The rational in the Introduction was revised as suggested.

Comment

4.Study design is mentioned in caption but not written anywhere. Kindly add name of the study design used.

Response

The name of study design was added to the Method section

Comment

5.Conclusion may kindly be strengthened.

Response

The Conclusion section was revised as suggested.

Reviewer #3:

Comment

The manuscript addresses an important and timely topic, focusing on medical students’ awareness and attitudes toward the health impacts of climate change, particularly in the underrepresented MENA region. The methodology is robust, with a clearly defined study population, ethical approval, and validated data collection tools. The discussion ties the findings to relevant global and regional studies, emphasizing the need for integrating climate change into medical curricula.

However, the same suggestions for improvement is to address typographical and grammatical errors such as in introduction Sentence: "Phenomena associated with climate change are plenty and include extreme weather (heat waves, cold weaves, and floods)..."Issue: "Cold weaves" should be "cold waves."

Response

Thank you. The indicated sentence was revised as indicated

Comment

In methods, Sentence: "Before distributing the questionnaire, the instrument was reviewed by two experts in the field to verify its clarity and content. "Issue: No grammatical error, but "clarity and content" could be expanded for specificity. Suggestion: "Before distributing the questionnaire, the instrument was reviewed by two experts in the field to verify its clarity, relevance, and content accuracy."

Response

The indicated sentence was revised as suggested.

Comment

In results, Sentence: "The percentages of male and female students were comparable (47.1% and 52.8% respectively)."Issue: A comma is missing after "52.8%."

Response.

The sentence was revised as suggested.

Sincerely yours,

Dr. Rami Saadeh

---

## [Decision Letter · Decision Letter 2]

15 Apr 2025

PONE-D-24-44537R2Knowledge and attitudes of medical students towards the health impact of climate change: a study from JordanPLOS ONE

Dear Dr. Saadeh,

Thank you for submitting your manuscript to PLOS ONE. After careful consideration, we feel that it has merit but does not fully meet PLOS ONE’s publication criteria as it currently stands. Therefore, we invite you to submit a revised version of the manuscript that addresses the points raised during the review process.

We look forward to receiving your revised manuscript.

Kind regards,

Fatma Refaat Ahmed, Ph.D.

Academic Editor

PLOS ONE

Journal Requirements:

Reviewers' comments:

Reviewer's Responses to Questions

**Comments to the Author**

1. If the authors have adequately addressed your comments raised in a previous round of review and you feel that this manuscript is now acceptable for publication, you may indicate that here to bypass the “Comments to the Author” section, enter your conflict of interest statement in the “Confidential to Editor” section, and submit your "Accept" recommendation.

Reviewer #2: All comments have been addressed

Reviewer #4: All comments have been addressed

2. Is the manuscript technically sound, and do the data support the conclusions?

Reviewer #2: Yes

Reviewer #4: Yes

3. Has the statistical analysis been performed appropriately and rigorously? 

Reviewer #2: I Don't Know

Reviewer #4: Yes

4. Have the authors made all data underlying the findings in their manuscript fully available?

Reviewer #2: Yes

Reviewer #4: Yes

5. Is the manuscript presented in an intelligible fashion and written in standard English?

Reviewer #2: No

Reviewer #4: Yes

6. Review Comments to the Author

Reviewer #2: Dear Author, language may kindly be further improved and refined.

xxxxxxxxxxxxxThanks xxxxxxxxxxxxxx

Reviewer #4: The study addresses an important public health issue and fills a gap in the MENA region regarding the awareness of climate change impacts on health among medical students.

There are typographical and grammatical issues throughout, such as “cold weaves” instead of “cold waves” and inconsistent use of singular/plural nouns. I suggest proofreading 'optional'

7. PLOS authors have the option to publish the peer review history of their article (what does this mean? ). If published, this will include your full peer review and any attached files.

**Do you want your identity to be public for this peer review?** For information about this choice, including consent withdrawal, please see our Privacy Policy .

Reviewer #2: No

Reviewer #4: **Yes: ** Nasser M Alorfi

---

## [Author Response · Author response to Decision Letter 3]

28 Apr 2025

Dear Editor of PLOS-ONE

We are thankful to the reviewers for their thoughtful and detailed review of our manuscript (ONE-D-24-44537 R2) entitled “Knowledge and attitudes of medical students towards the health impact of climate change: a study from Jordan”. All the recommended changes and edits have been incorporated in the revised version of our manuscript. We greatly appreciate the time and effort invested to produce this detailed review. Our point-by-point response is provided below.

Journal Requirements:

Response

The references were checked and formatted according to PLOS style. They are complete and correct. None of the cited papers have been retracted.

Reviewers' comments:

Reviewer #2: Dear Author, language may kindly be further improved and refined.

xxxxxxxxxxxxxThanks xxxxxxxxxxxxxx

Reviewer #4: The study addresses an important public health issue and fills a gap in the MENA region regarding the awareness of climate change impacts on health among medical students.

There are typographical and grammatical issues throughout, such as “cold weaves” instead of “cold waves” and inconsistent use of singular/plural nouns. I suggest proofreading 'optional'

Response

The manuscript was checked for English. Changes were highlighted in the revised manuscript.

Sincerely yours,

Corresponding author

---

## [Editor Report · Decision Letter 3]

4 May 2025

Knowledge and attitudes of medical students towards the health impact of climate change: a study from Jordan

PONE-D-24-44537R3

Dear Dr. Saadeh,

We’re pleased to inform you that your manuscript has been judged scientifically suitable for publication and will be formally accepted for publication once it meets all outstanding technical requirements.

Kind regards,

Fatma Refaat Ahmed, Ph.D.

Academic Editor

PLOS ONE
---

## [Editor Report · Acceptance letter]

PONE-D-24-44537R3

PLOS ONE

Dear Dr. Saadeh,

I'm pleased to inform you that your manuscript has been deemed suitable for publication in PLOS ONE. Congratulations! Your manuscript is now being handed over to our production team.

Kind regards,

on behalf of

Dr. Fatma Refaat Ahmed

Academic Editor

PLOS ONE